# American Masters Road Running Records—The Performance Gap Between Female and Male Age Group Runners from 5 Km to 6 Days Running

**DOI:** 10.3390/ijerph16132310

**Published:** 2019-06-29

**Authors:** Caio Victor Sousa, Samuel da Silva Aguiar, Thomas Rosemann, Pantelis Theodoros Nikolaidis, Beat Knechtle

**Affiliations:** 1Graduate Program in Physical Education, Catholic University of Brasília, 71966-700 Brasília, DF, Brazil; 2Miller School of Medicine, University of Miami, Miami, FL 33136, USA; 3Physical Education Department, University Center-UDF, 71966-700 Brasilia, DF, Brazil; 4Institute of Primary Care, University of Zurich, 8091 Zurich, Switzerland; 5Exercise Physiology Laboratory, 18450 Nikaia, Greece; 6Medbase St. Gallen Am Vadianplatz, 9001 St. Gallen, Switzerland

**Keywords:** athlete, running, ultra-endurance, endurance, marathon

## Abstract

Recent studies investigating elite and master athletes in pool- and long-distance open-water swimming showed for elite swimmers that the fastest women were able to outperform the fastest men, and for master athletes that elderly women were able to achieve a similar performance to elderly men. The present study investigating age group records in runners from 5 km to 6 days aimed to test this hypothesis for master runners. Data from the American Master Road Running Records were analyzed, for 5 km, 8 km, 10 km, 10 miles, 20 km, half-marathon, 25 km, 30 km, marathon, 50 km, 50 miles, 100 km, 100 miles, 12 h, 24 h, 48 h and 144 h, for athletes in age groups ranging from 40 to 99 years old. The performance gap between men and women showed higher effects in events lengthening from 5 km to 10 miles (d = 0.617) and lower effects in events lengthening from 12 to 144 h (d = 0.304) running. Both other groups showed similar effects, being 20 km to the marathon (d = 0.607) and 50 km to 100 miles (d = 0.563). The performance gap between men and women showed higher effects in the age groups 85 years and above (d = 0.953) followed by 55 to 69 years (d = 0.633), and lower effects for the age groups 40 to 54 years (d = 0.558) and 70 to 84 years (d = 0.508). In summary, men are faster than women in American road running events, however, the sex gap decreases with increasing age but not with increasing event length.

## 1. Introduction

Sport performance differences between men and women have been previously reported in several different modalities, such as in open-water swimming, in marathon running and in Ironman triathlon [1,2,3,4]. The most likely explanation of the sex gap in sports is possibly due to different human skeletal muscle gene expression and their interaction with sex-specific hormones [5,6]. This fact is leading to larger muscles and more functional muscle fibers in men, which elicits greater muscle strength/power and/or endurance [7,8,9].

However, in endurance sports the sex gap has been changing over the last decades [10]. For instance, for elite athletes, Nikolaidis et al. [4] reported that women were faster than men in open-water swimming in the English Channel Crossing. Furthermore, in long-distance pool swimming [11] and in open-water swimming like the English Channel Crossing [12] women were able to achieve a similar performance to men. In some ultra-distance swimming events, women were even able to outperform men [4]. Moreover, for other sports disciplines elite women seemed to have improved more than men in marathon running [2] and in Ironman triathlon [3]. Another report showed that the sex gap decreased with increasing age in marathon runners [1]. And in ultra-endurance road races, women seem to have a higher range of peak performance than men (30–54 years vs. 30–49 years) but the peak of age-related performance decline was similar between them (60–64 years) [13].

For master athletes, it seems that elderly women can reach the performance of elderly men, especially in pool and open-water swimming. Recent studies investigating master pool swimmers in freestyle [14], backstroke [15], butterfly [16], breaststroke [17], individual medley [14], and open-water [18] swimming showed that women in the older age groups (i.e., older than 75 years) achieved a similar performance to men. 

Men have shown to have better values of physiological performance determinants such as higher VO_2_max, lower external load at anaerobic threshold, and better running economy [19]. However, women normally accumulate more body fat and also have an increased efficiency to produce energy through oxidation of this substrate during endurance performance [20], which could be an advantage in events with a longer length. Furthermore, the natural decline in sex-specific hormones in men while aging [21] may affect their performance, possibly leading to the recently observed sex difference reduction as athletes age [1].

Aging is a natural, progressive and inevitable biological process [22]. As humans age, a decrease in anabolic hormones may lead to the well-known age-related aspects such as sarcopenia, osteopenia, increased body fat, and loss of function in several other tissues [22]. Although master athletes have a good life-style capable of improved health and attenuated aging [23,24], they are not immune to the biological age-related effects, which may reflect in a performance decrease while aging. 

We therefore aimed to investigate the possible effect of age and race length in endurance American master road runners of different lengths, which to the best of our knowledge has not yet been investigated for age group runners. We hypothesized, based on recent findings for elite and master pool- and long-distance swimmers, that the longer the length and/or higher the age the smaller the sex difference would be. 

## 2. Materials and Methods 

To test our hypothesis, the world records in each distance and age-group of both men and women were included. Dispersion plots with linear regression analysis may show if the performance pattern of men is different from women with increasing age and increasing race length. 

### 2.1. Ethical Approval

This study was approved by the Institutional Review Board of Kanton St. Gallen, Switzerland, with a waiver of the requirement for informed consent of the participants as the study involved the analysis of publicly available data (EKSG 01/06/2010). 

### 2.2. Data 

All data were the official USA Track and Field race records from master athletes, publicly available from their web site [25]. The American records from all road running events for men and women were collected for further analysis. Collection of data ranged the years of 1970 to 2017. The last time the database was checked for new records was March 2019. The included events were: 5 km, 8 km, 10 km, 10 miles, 20 km, half-marathon, 25 km, 30 km, marathon, 50 km, 50 miles, 100 km, and 100 miles for distance-limited races, and 12 h, 24 h, 48 h and 144 h for time-limited races (for all age-groups). Data included age groups in 5-year-intervals ranging from 40–44 to 95–99 years old (for all events). The data sample included 214 men and 200 women (i.e., total sample size *n* = 414). Race times in the time-limited races and achieved distance in the distance-limited races were all converted to running speed for results comparison.

### 2.3. Statistical Analysis 

Data analysis was conducted with pooled groups by events, pooled groups by age-group and without any grouping. An analysis of variance adjusted by age (ANCOVA) was applied to compare men and women performances when pooled by event or age group. Furthermore, the effect size [26] between men and women was calculated for each event and age group and then pooled in different groups. Finally, linear regressions with individual values were conducted between relative sex difference (%) and age or event length (km). The significance level was 5% (*p* < 0.05). All procedures were performed using Statistical Software for the Social Sciences (IBM, SPSS v21.0, Chicago, IL, USA) and GraphPad Prism (Graph Pad Prism v7.0, San Diego, CA, USA).

## 3. Results

The comparisons of running speed between men and women were significantly different in all endurance events for all age groups. Men have a significantly faster average running speed in endurance events lengthening from 5 km to 144 h running (Table 1). 

The comparisons between men and women were not statistically different in almost every age group for all events. Men have a significantly faster average running speed in the age group 60–64 years (Table 2). 

The performance gap between men and women showed higher effects in events lengthening from 5 km to 10 miles (d = 0.617) and lower effects in events lengthening from 12 to 144 h (d = 0.304) running (Figure 1A). Both other groups showed similar effects, being from 20 km to the marathon (d = 0.607) and from 50 km to 100 miles (d = 0.563). The performance gap between men and women showed higher effects in the age groups 80–85 years and above (d = 0.953) followed by 55–59 to 65–69 years (d = 0.633), and the lower effects for the age groups 40–44 to 50–54 years (d = 0.558) and 70 to 84 years (d = 0.508) (Figure 1B).

Dispersion data of ‘performance sex difference ratio’ vs. ‘age’ was plotted in Figure 2 for the linear regression analysis. Linear regressions showed an increase in the sex gap with increasing age for all endurance events excepts 144 hours running, with data only available for age groups up to 60–64 years old (Figure 2). 

Dispersion data of ‘performance sex difference ratio’ vs. ‘event length’ was plotted in Figure 3 for the linear regression analysis. Linear regressions with sex gap and event length did not present a pattern of an increase or a decrease with increasing distance in different age groups (Figure 3). Age groups 60–64, 75–79 and 80–84 years showed an increasing sex gap with increasing distance and/or duration, whereas age groups 45–49, 50–54, 70–74 and 85–89 years showed a decreasing sex gap with increasing distance and/or duration. 

## 4. Discussion

In this study we hypothesized to find that women would be able to close the gap to men in higher ages and longer race distances as has been recently reported for pool- and open-water swimmers. The main finding of this investigation was that the sex gap in sports performance for American road runners decreased with increasing age but did not seem to change with increasing distance and/or duration. Nonetheless, men remained faster than women regardless of age and race distance and/or duration.

The overall sex difference (men outscored women) in running performance in the examined distances (5 km–6 days) might reflect the corresponding sex differences in the physiological determinants of performance in these events, i.e., maximal oxygen uptake, anaerobic threshold and running economy [27]. For instance, men marathon runners had higher VO_2max_ than their women counterparts [28]. On the other hand, women had more economical running due to their smaller body mass [29].

Every human suffers from a natural, constant and inevitable physiological decline while aging [22], which is a main cause for the reduced performance in older age. However, at a certain age (i.e., between 65 and 75 years) men go through a substantial decrease in sex-specific hormones that plays a fundamental role in skeletal muscle mass maintenance, a phase popularly known as andropause [21,30]. 

Indeed, it has been previously reported that male master track-and-field athletes showed a substantial decrease in performance at around that age [31], most likely due to the expressive reduction of type II muscle fibers [32]. Since women undergo substantial reductions in sex hormones which are not that fundamental to performance as for men, it is reasonable to infer that the sex gap closing with increasing age is a consequence of a significant reduction in men’s performance after andropause. However, since no data for hormones was collected in the present study, it can only be inferred that it might lead to higher performance in men.

Men have more skeletal muscle mass and muscle functionality due to a higher secretion of sex-specific hormones and gene expressions [5,6,7,8,9], which leads to their increased performance in comparison to women, as reported in the present article. However, the better ability to achieve higher peak power output also causes men to have a greater fatigability [9] which may mean that, in theory, women would be better in longer events.

Furthermore, women show greater rates of fat oxidation at sub-maximal intensities of exercise [20,33], which could preclude them to accumulate significant quantities of metabolic waste that leads to fatigue [9]. The lower fatigability and increased rate of fat oxidation are powerful tools for greater performances in endurance and ultra-endurance events longer than the classical marathon distance. Nevertheless, it does not seem to reflect in records performance in American road runners, since it is not possible to see a pattern for all age groups of decreasing/increasing sex difference with increasing race distance and/or duration. 

We hypothesized that women in long-distance running would close the gap to men in older age groups as it has been shown in freestyle [14], backstroke [15], butterfly [16], breaststroke [17], individual medley [14], and open-water [18] swimming. However, in contrast to swimming, we did not find the same pattern in running. A very likely explanation is the difference in the locomotion mode (i.e., non-weight bearing in swimming compared to weight bearing in running) and the longer performance times in running compared to pool swimming. Another potential explanation is that in the analysis of the swimming events all recorded athletes in each age group were included in the data analysis, leading to very large sample sizes in each age group, whereas in this analysis of record performances only the records (i.e., the fastest women and men for each distance/length and age group) were considered.

Since this study only included American road running records, the results should be viewed with caution before extrapolating them to other populations. Although the American road master events are open to everyone, only Americans are eligible for an American record. Moreover, master athletes are an elite aging population with unique life-style habits that may distinguish them from the general population [34]. However, the singularity of the sample is also a strength of the present manuscript, since data from elite athletes (masters or not) are always challenging. It is also important that some of the linear regressions had some points that do not seem to fit the regression, such as half-marathon (R^2^ = 0.336), 100 miles (R^2^ = 0.354), 24 h (R^2^ = 0.437), 48 h (R^2^ = 0.296), 144 h (R^2^ = 0.099). All others were above 50% (R^2^ > 0.5), and the highlighted regressions should be viewed with caution. 

## 5. Conclusions

In conclusion, men were faster than women in American Masters road running events. However, the sex gap decreased with increasing age but not with increasing distance and/or duration. Future research should investigate this aspect with master athletes running the same events but include all participants. An analysis across the decades may also provide valuable and novel information.

## 6. Practical Applications 

Present results showed that an increasing race length in endurance events do not reduce the performance gap between men and women, but an increasing age does. It is reasonable to infer that the age-related decline of sex-specific hormones affects men with increasing age, and it might play a role between men and women with increasing race length. Men master athletes have a greater performance decline than women with increasing age and should address this issue with extra care in proper nutrition and training, and consider annual hormones check-ups to adjust substantial changes, regardless of their race specialization (5 km to ultra-marathons). Although women have a more subtle performance decline with increasing age than men, the natural effects of aging also affect them, and a more substantial performance decline may be expected from those with better performance.

## Figures and Tables

**Figure 1 ijerph-16-02310-f001:**
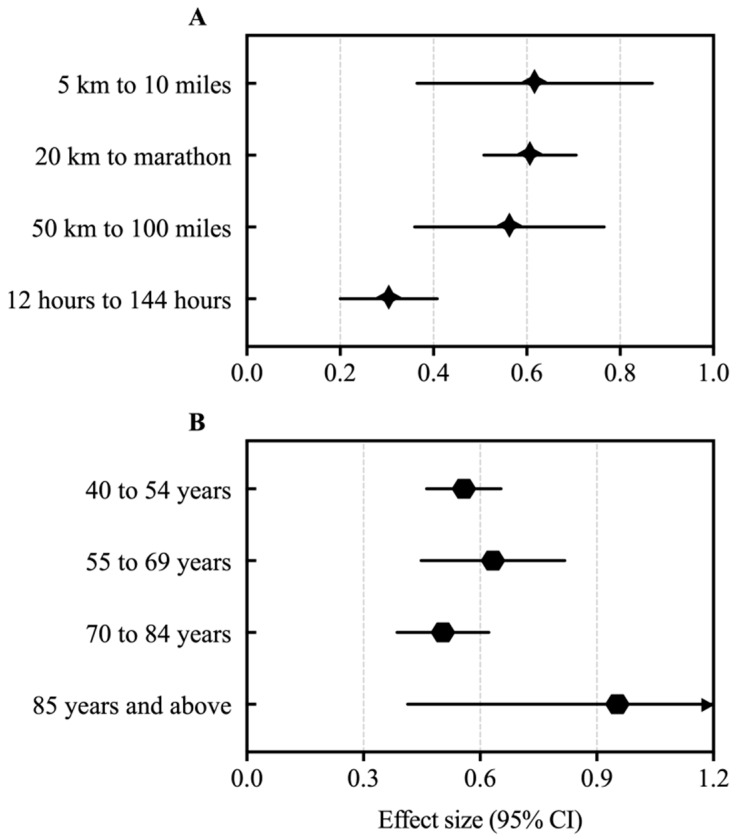
Performance gap between men and women in American masters road running records. CI: confidence interval. Greater the effect size (>0), greater the difference between men and women.

**Figure 2 ijerph-16-02310-f002:**
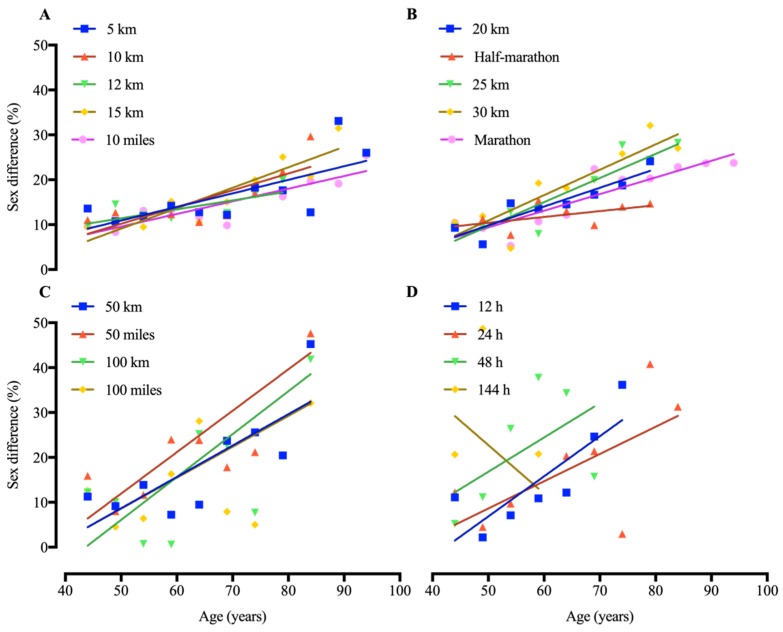
Linear regression between performance sex difference (%) and age in American masters road running records. With the positive slopes, higher the age, higher the sex difference (%) in each event. (**A**) events 5 km to 10 miles; (**B**) 20 km to Marathon; (**C**) 50 km to 100 miles; (**D**) 12 to 144 h.

**Figure 3 ijerph-16-02310-f003:**
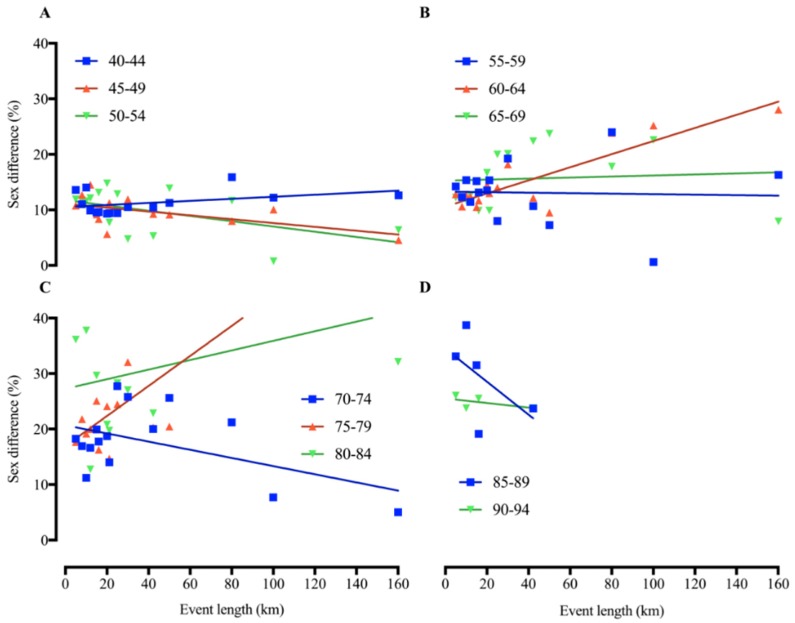
Linear regression between performance sex difference (%) and event length (km) in America masters road running records. (**A**) Age groups from 40 to 54 years old; (**B**) 55 to 69 years old; (**C**) 70 to 84 years old; (**D**) 85 to 94 years old.

**Table 1 ijerph-16-02310-t001:** Average running speed (km·h^−1^) of the American masters road running records by event. Data expressed as mean and standard deviation (±).

Events	Men	Women	*p*-Value
5 km (*n* = 22)	15.5 ± 5.2	13.4 ± 4.4	<0.00001
8 km (*n* = 18)	15.9 ± 3.7	14.5 ± 2.7	<0.00001
10 km (*n* = 18)	14.9 ± 4.2	13.7 ± 3.2	0.00009
12 km (*n* = 16)	15.9 ± 3.2	14.8 ± 2.1	0.00012
15 km (*n* = 20)	15.7 ± 4.2	10.0 ± 3.1	0.00041
10 miles (*n* = 22)	15.0 ± 4.1	13.0 ± 3.6	0.00037
20 km (*n* = 16)	16.2 ± 2.6	14.3 ± 2.8	<0.00001
Half-marathon (*n* = 16)	15.5 ± 4.4	12.6 ± 3.0	0.00003
25 km (*n* = 18)	15.1 ± 3.2	13.2 ± 3.2	0.00001
30 km (*n* = 18)	14.0 ± 4.0	12.6 ± 3.5	0.00002
Marathon (*n* = 22)	14.1 ± 4.0	12.0 ± 3.6	<0.00001
50 km (*n* = 18)	12.9 ± 3.3	11.0 ± 3.3	0.00047
50 miles (*n* = 18)	12.0 ± 2.5	9.6 ± 3.4	<0.00001
100 km (*n* = 18)	11.2 ± 2.8	8.7 ± 3.8	0.00015
100 miles (*n* = 18)	8.0 ± 3.2	7.6 ± 2.7	0.01730
12 h (*n* = 14)	9.2 ± 2.3	8.9 ± 2.3	0.00052
24 h (*n* = 18)	8.2 ± 2.3	7.4 ± 2.3	0.00009
48 h (*n* = 12)	5.8 ± 1.7	5.0 ± 2.0	0.00023
144 h (*n* = 8)	4.6 ± 1.4	4.3 ± 1.0	0.01795

*p*-value: univariate model adjusted by age; Sample size was equal for men and women.

**Table 2 ijerph-16-02310-t002:** Average running speed (km·h^−1^) of the American masters road running records by age-group. Data expressed as mean and standard deviation (±).

Age Groups	Men	Women	*p*-Value
40–44 years (*n* = 36)	17.4 ± 4.3	14.5 ± 4.4	0.20427
50–54 years (*n* = 36)	16.3 ± 4.2	13.7 ± 4.3	0.07979
55–59 years (*n* = 36)	15.0 ± 3.9	13.6 ± 3.5	0.31009
60–64 years (*n* = 36)	15.5 ± 3.9	12.1 ± 3.8	0.00368
65–69 years (*n* = 34)	14.3 ± 3.4	12.0 ± 3.4	0.05812
70–74 years (*n* = 38)	12.8 ± 3.9	11.7 ± 3.0	0.26013
75–79 years (*n* = 32)	12.2 ± 3.7	10.7 ± 2.8	0.11431
80–84 years (*n* = 30)	11.3 ± 3.7	8.8 ± 3.4	0.29139
85–89 years (*n* = 10)	8.7 ± 3.9	7.3 ± 3.1	0.20891
90–94 years (*n* = 6)	7.9 ± 2.6	7.1 ± 0.7	0.42614
95–99 years (*n* = 4)	7.1 ± 1.3	5.3 ± 1.1	0.04698

*p*-value: univariate model adjusted by age. Sample size was equal for men and women.

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
