# Peer review of "American Masters Road Running Records—The Performance Gap Between Female and Male Age Group Runners from 5 Km to 6 Days Running"

_ijerph, 2019, doi:10.3390/ijerph16132310_

Round 1

Reviewer 1 Report

In this manuscript the authors analysed the records of road running in American Masters events for comparison between genders in age range of 40 to 99 years. It was concluded that men are faster than women in these events, but the gap decreases with aging but not with the event length. The topic is interesting. However, there are a number of areas in the manuscript that require more information or improvement of clarity, as shown below.

Line 24: the age range was 40-94 here, but it was 40-99 in Line 83.

Lines 57-63: The rationale to support the hypothesis on the possible effect of age on endurance performance could be strengthened. The gender differences in body composition, muscle quality and quantity, and fat metabolism were mentioned, but how age would affect these (and other) factors should be discussed.

Line 77: more details about the data would be helpful, e.g. in which year the website of records was accessed and the records were set in which year (or range of years). Many factors may change over time, such as environment of the competitions, training methods, nutrition, etc. that all may contribute to the performance changes.

Lines 84-86: The averaging speed (as presented in Table 1) should be clearly defined here. If only the best performance (event record) was used in the analysis, how the average and SD values for each event were obtained (as presented in Table 1)? E.g. were the average and SD values for all age groups in that event? If so, please consider add “for all age groups” at the end of Line 101; and “for all events” at the end of Line 106.

The meaning of panel A, B, C and D should be explained in the text or title of Figures 2 and 3, e.g. performance by distance or age.

It seems some records are not available for every event and or for every age group (as shown in the Figures). It would be more informative to readers (to have a better understanding of the stats) if the number in each group were presented in Table 1 and 2.

In Figure 2 it is difficult to identify the information on the relationship between the performance difference and age (to support the conclusion), particularly in panels C and D where appear to be some outliers at various ages and the linear regression may not fit the data well. Is that possible to add a hypothetical regression line for no change in the gender gap? Similar problem is seen with Figure 3.

Line 141: a less definite word should be used in speculation of the underlying mechanisms, e.g. “might reflect”, because no such data was collected from the athletes involved in this study.

Line 158: again, no data was collected for hormones and gene expressions in this study, so consider use “which might lead to their higher performance……”.

Line 159: consider use “the better ability…..” instead of “the increased ability……”.

Line 179: this is confusing, as the data for 95-99 years old runners were presented in Table 2.

Lines 180-184: the readers would appreciate more comprehensive background information, such as whether the American Master games only open to American citizens or they are international events.

Line 186: consider add “masters” in the first sentence.

Author Response

REVIEWER 1

INITIAL COMMENT: In this manuscript the authors analysed the records of road running in American Masters events for comparison between genders in age range of 40 to 99 years. It was concluded that men are faster than women in these events, but the gap decreases with aging but not with the event length. The topic is interesting. However, there are a number of areas in the manuscript that require more information or improvement of clarity, as shown below.

COMMENT #1: Line 24: the age range was 40-94 here, but it was 40-99 in Line 83.

RESPONSE #1: The sentence was corrected. Thank you for the comment.

COMMENT #2: Lines 57-63: The rationale to support the hypothesis on the possible effect of age on endurance performance could be strengthened. The gender differences in body composition, muscle quality and quantity, and fat metabolism were mentioned, but how age would affect these (and other) factors should be discussed.

RESPONSE #2: We agree with the expert reviewer. A new paragraph discussing how age may affects performance was added. See changes in text marked as red and the added text below.

[…]

“Aging is a natural, progressive and inevitable biological process [23]. As humans age, a decrease in anabolic hormones may lead to the well-known age-related aspects such as, sarcopenia, osteopenia, increased body fat and loss of function in several other tissues [23]. Although master athletes have a good life-style capable of improved health and attenuated aging [24,25], they are not immune to the biological age-related effects, which may reflect in a performance decrease while aging.”

[…]

COMMENT #3: Line 77: more details about the data would be helpful, e.g. in which year the website of records was accessed and the records were set in which year (or range of years). Many factors may change over time, such as environment of the competitions, training methods, nutrition, etc. that all may contribute to the performance changes.

RESPONSE #3: The suggestion is very pertinent and the information required was added within the Methods section. See all changes highlighted in red.

COMMENT #4: Lines 84-86: The averaging speed (as presented in Table 1) should be clearly defined here. If only the best performance (event record) was used in the analysis, how the average and SD values for each event were obtained (as presented in Table 1)? E.g. were the average and SD values for all age groups in that event? If so, please consider add “for all age groups” at the end of Line 101; and “for all events” at the end of Line 106.

RESPONSE #4: Yes, the analysis was performed as you described. We added the phrases at the end of the sentences where you suggested.

COMMENT #5: The meaning of panel A, B, C and D should be explained in the text or title of Figures 2 and 3, e.g. performance by distance or age.

RESPONSE #5: The meaning each panel was explained in the respective figure legend, and the meaning of the figure was included in the text. Thank you for the comment.

COMMENT #6: It seems some records are not available for every event and or for every age group (as shown in the Figures). It would be more informative to readers (to have a better understanding of the stats) if the number in each group were presented in Table 1 and 2.

RESPONSE #6: You are correct. Some age-groups and events have inconsistent data for either men or women, an if one of these values were missing/inconsistent, the age-group of that particular event were excluded. We added the sample sizes for each group in both Tables, as suggested. Thank you for the comment.

COMMENT #7: In Figure 2 it is difficult to identify the information on the relationship between the performance difference and age (to support the conclusion), particularly in panels C and D where appear to be some outliers at various ages and the linear regression may not fit the data well. Is that possible to add a hypothetical regression line for no change in the gender gap? Similar problem is seen with Figure 3.

RESPONSE #7: We agree with the expert reviewer regarding the importance of highlight the regressions with too many points that do not fit the line. Although we cannot add a hypothetical regression line, we presented the R2 value for the regression with fitting percentage were below 50%. We included a statement within the limitations of the study. Please see changes highlighted in red and below.

[…] “It is also important that some of the linear regressions had some points that do not seem to fit the regression such as, half-marathon (R2 = 0.336), 100 miles (R2 = 0.354), 24hr (R2 = 0.437), 48hr (R2 = 0.296), 144hr (R2 = 0.099). All other were above 50% (R2 > 0.5), and the highlighted regressions should be view with caution.”

[…]

COMMENT #8: Line 141: a less definite word should be used in speculation of the underlying mechanisms, e.g. “might reflect”, because no such data was collected from the athletes involved in this study.

RESPONSE #8: We agree with the reviewer and changed the sentence, as suggested.

COMMENT #9: Line 158: again, no data was collected for hormones and gene expressions in this study, so consider use “which might lead to their higher performance……”.

RESPONSE #9: We agree with the expert reviewer and added a new sentence as suggested. Please see changes highlighted in red.

COMMENT #10: Line 159: consider use “the better ability…..” instead of “the increased ability……”.

RESPONSE #10: changed as suggested.

COMMENT #11: Line 179: this is confusing, as the data for 95-99 years old runners were presented in Table 2.

RESPONSE #11: The expert reviewer is correct. The sentence was removed. Thank you for the comment.

COMMENT #12: Lines 180-184: the readers would appreciate more comprehensive background information, such as whether the American Master games only open to American citizens or they are international events.

RESPONSE #12: We agree with the expert reviewer and added this information before the conclusion, as suggested.

COMMENT #13: Line 186: consider add “masters” in the first sentence.

RESPONSE #13: It was added, as suggested.

Reviewer 2 Report

This. study is about the development of running time with age. Especially the difference between genders has been assessed. While there is indeed some merit in this type of studies, there are some points that should be addressed.

Mayor comments;

- the statistical power is not clear. Please include in the tables n for each group. 

-it is not clear how the regressions were calculated. Was it an pooled approach too? ‘if only 3 data points are available, the error can be very high. (Fig 2, d)

- The authors include too much their own work about swimming, please provide comparison to running. A comparison to other sports is also reasonable, but then it should be over many different sports. 

minor points:

- There are many irregularities in the data that should be further discussed. Why ist he running speed for 20 km so much higher than for half marathon?

-please check the affiliations a d the email addresses. 

Author Response

REVIEWER 2

INITIAL COMMENT: This. study is about the development of running time with age. Especially the difference between genders has been assessed. While there is indeed some merit in this type of studies, there are some points that should be addressed.

COMMENT #1: the statistical power is not clear. Please include in the tables n for each group. 

RESPONSE #1: The sample size for each group (n) was added in both tables, as suggested.

COMMENT #2: it is not clear how the regressions were calculated. Was it an pooled approach too? ‘if only 3 data points are available, the error can be very high. (Fig 2, d)

RESPONSE #2: The regressions were calculated with individual values, there are no dispersion values for each point. This was highlighted in red within the text, please see a copy below.

[…] groups. Finally, linear regressions with individual values were conducted between relative sex difference (%) and age or event length (km). The significance [...]

COMMENT #3: - The authors include too much their own work about swimming, please provide comparison to running. A comparison to other sports is also reasonable, but then it should be over many different sports. 

RESPONSE #3: We agree with the expert reviewer and removed the non-essential swim studies and added more work with running. Please see all changes highlighted in red.

COMMENT #4: There are many irregularities in the data that should be further discussed. Why ist he running speed for 20 km so much higher than for half marathon?

RESPONSE #4: Road running races usually takes place in outdoors courses with different characteristics that could increase or decrease a race average velocity (i.e. altimetry, weather, sharp curves). Furthermore, since the half-marathon is more popular than 20km race, it is possible that some less easy races may have pull down the mean running speed. Either way, we double checked the data in the source for inconsistencies.

COMMENT #5: please check the affiliations a d the email addresses. 

RESPONSE #5: Checked and corrected, as suggested.

Round 2

Reviewer 1 Report

The authors have address my previous queries properly. I have no further major comments.

A few minor points that need authors’ attention:

Line 20: consider add a coma after “men”.

Line 109 (and Line 119): use “Masters”.

Line 111 (also Line 121): use lower case for “sample” and add “.” at the end.

Line 112: delete “all”.

Line 214: delete “s” from “reduces”.

Line 216: “but it to play” does not make sense. Not sure what the authors wanted to say.

Line 217-8: consider move the “,” from after “athletes” to after “age”.

Line 221: should the “then” be “them”?

Please do a thorough proof reading.

Reviewer 2 Report

thank you for addressing the comments and suggestions. well done.